# Different Sensitivity of Advanced Bronchial and Alveolar Mono- and Coculture Models for Hazard Assessment of Nanomaterials

**DOI:** 10.3390/nano13030407

**Published:** 2023-01-19

**Authors:** Elisabeth Elje, Espen Mariussen, Erin McFadden, Maria Dusinska, Elise Rundén-Pran

**Affiliations:** 1Health Effects Laboratory, Department for Environmental Chemistry, NILU—Norwegian Institute for Air Research, 2007 Kjeller, Norway; 2Department of Molecular Medicine, Institute of Basic Medical Sciences, University of Oslo, 0372 Oslo, Norway; 3Department of Air Quality and Noise, Norwegian Institute of Public Health, 0456 Oslo, Norway

**Keywords:** NAMs—new approach methodologies, ALI—air–liquid interface, genotoxicity, BEAS-2B, A549, NM-300K, DNA damage, chromosomal damage, cytokines

## Abstract

For the next-generation risk assessment (NGRA) of chemicals and nanomaterials, new approach methodologies (NAMs) are needed for hazard assessment in compliance with the 3R’s to reduce, replace and refine animal experiments. This study aimed to establish and characterize an advanced respiratory model consisting of human epithelial bronchial BEAS-2B cells cultivated at the air–liquid interface (ALI), both as monocultures and in cocultures with human endothelial EA.hy926 cells. The performance of the bronchial models was compared to a commonly used alveolar model consisting of A549 in monoculture and in coculture with EA.hy926 cells. The cells were exposed at the ALI to nanosilver (NM-300K) in the VITROCELL^®^ Cloud. After 24 h, cellular viability (alamarBlue assay), inflammatory response (enzyme-linked immunosorbent assay), DNA damage (enzyme-modified comet assay), and chromosomal damage (cytokinesis-block micronucleus assay) were measured. Cytotoxicity and genotoxicity induced by NM-300K were dependent on both the cell types and model, where BEAS-2B in monocultures had the highest sensitivity in terms of cell viability and DNA strand breaks. This study indicates that the four ALI lung models have different sensitivities to NM-300K exposure and brings important knowledge for the further development of advanced 3D respiratory in vitro models for the most reliable human hazard assessment based on NAMs.

## 1. Introduction

The production and usage of nanomaterials (NMs) are rising, increasing the risk of human exposure. Inhalation is the most important exposure route for airborne nanomaterials (NMs) and particulate matter (PM) in humans, making the respiratory system a first-target organ [1]. The respiratory tract consists of the tracheobronchial region leading into the alveolar region, where gas exchange with blood occurs across the thin lung–blood barrier (0.4 µm) [1,2]. Besides gas exchange, a main function of the lower respiratory tract is defense against inhaled toxicants [1]. Interaction with and deposition of inhaled NMs are likely to occur in the bronchial and alveolar region. Particle deposition is dependent upon the NMs’ physicochemical properties, such as size and solubility [2].

NMs and their dissolved compounds can cause primary effects in the respiratory system, or secondary circulatory effects after crossing the lung–blood barrier and taken up in the blood. A human study has shown the translocation of inhaled gold NM or its dissolved species into the circulatory system and accumulation at sites of vascular disease [3]. Gold was detected in blood and urine up to three months after inhalation exposure to gold NMs [3,4]. Translocation of silver NMs has been seen in in vivo studies in rodents [5].

In order to comply with the 3R´s principle to reduce, refine and replace animal experiments, new advanced in vitro models are developed to better simulate the complexity of human lungs. Reliable in vitro models of the airway system are of critical importance for the risk assessment and governance of NMs and other environmental pollutants [6,7]. Human cells cultured on a microporous membrane at the air–liquid interface (ALI) with cell culture medium only at the basolateral side, represent a highly relevant model for inhalation toxicity studies [8]. Human lung cell lines such as A549 and BEAS-2B are commonly used as model cells in respiratory toxicology. A549 cells are alveolar type-II carcinoma cells, while BEAS-2B cells are immortalized cells from normal human bronchial epithelia. Both A549 and BEAS-2B cells form monolayers when cultivated at the ALI [9,10]. In order to further advance the models, cocultures with other cell types, such as macrophages, dendritic cells, or endothelial cells, can be established. The ALI exposure model aims to better mimic the physiology of the respiratory system and is regarded as a more relevant in vitro model compared to submerged exposure. Aerosolized exposure to the particles on top of the cells introduces less changes in the physicochemical properties of the test substance compared with submerged exposure [8].

Inhalation exposure to NMs, PM or other compounds may lead to adverse human effects. Genotoxicity is a critical endpoint in the hazard assessment of chemicals, including NMs, and should be assessed both at the level of DNA/genes and chromosomes. The comet assay is a widely used assay for determining DNA damage as DNA strand breaks (SBs), and as oxidized or alkylated bases by the inclusion of a repair enzyme such as formamidopyrimidine DNA glycosylase (Fpg) [11]. For the detection of chromosomal damage, the most-used test is the micronucleus assay (OECD test guideline 487), which detects the formation of micronuclei from chromosomes, chromatid fragments or whole chromosomes that lag behind in cell division [12,13]. So far, a very limited number of studies have addressed several genotoxicity endpoints in ALI models. Our approach, combining advanced and more physiologically relevant in vitro respiratory models and exposure systems with genotoxicity testing (by both comet and micronucleus assays), will support the hazard characterization of NMs for risk assessment and safe use.

NMs can induce DNA damage by direct contact with DNA, or indirectly via NM-induced oxidative stress or intermediate molecules and processes in cells (primary genotoxicity). Secondary genotoxicity can be driven by an inflammatory response [14]. The airway epithelium is an integrated part of the inflammatory defense response after inhalation exposure to toxicants. Pro-inflammatory cytokines are considered biomarkers of NM-induced toxicity and can be linked with adverse effects. The pro-inflammatory cytokines IL-6 and IL-8 are among the cytokines predominately secreted by monocytes, and both are coupled to lung injury and considered biomarkers of lung disease [15,16,17]. IL-8 can also act as a chemokine [17]. The bronchial epithelium serves as a first-line defense system against inhaled pathogens mainly by the release of chemokines, such as IL-8 [18]. The cytokines IL-6 and IL-8 have been shown to be secreted by airway epithelial, including BEAS-2B cells, and endothelial cells and be involved in lung inflammation responses [18,19,20,21]. IL-6 has been shown to be released from BEAS-2B cells after exposure to particulate matter below 1 µm in size (PM1), and both IL-6 and IL-8 were induced in BEAS-2B cells after exposure to the PM2.5 fraction [22,23]. Endothelial EA.hy926 cells were shown to release IL-6 and IL-8 after exposure to silica NMs [19].

The A549 cell line has frequently been used in coculture lung models and has been shown to be useful in a range of applications for hazard assessment of NMs [24,25,26,27,28,29,30,31,32,33,34,35]. The non-cancerous origin of BEAS-2B cells may make the cell line more relevant for use in risk governance of NMs, particularly as a bronchial respiratory model. Coculture models with BEAS-2B in ALI conditions for hazard assessment are, however, much less characterized than those with A549. The main aim of this study was to characterize an advanced respiratory model with BEAS-2B bronchial cells cultivated in ALI models, after exposure to an aerosolized reference silver NM, NM-300K. The cells were cultivated both as monocultures and in cocultures with human endothelial EA.hy926 cells. Cells from ALI cultures were analyzed for cytokine secretion, cytotoxicity, barrier integrity, DNA damage by the comet assay and chromosomal damage by the cytokinesis-block micronucleus assay. Importantly, the responses obtained with the bronchial BEAS-2B model were compared with the A549 alveolar model. The experimental design brilliantly allows for the comprehensive analysis of several endpoints from the same sample, facilitating increased throughput, better comparability, reduced costs, and sustainability by design to support the development of new approach methodologies (NAMs) and next-generation risk assessment (NGRA) of NMs.

## 2. Materials and Methods

### 2.1. Experimental Design

An experimental design combining the analysis of several endpoints from the same sample was developed. The same inserts with cells at the ALI were used for the analysis of cytokine secretion in basolateral media (enzyme-linked immunosorbent assay, ELISA), Ag permeation (inductively coupled plasma mass spectrometry, ICP-MS), cell viability (alamarBlue assay), cell proliferation, DNA damage and oxidized base lesions (enzyme modified version of the comet assay), and chromosomal damage (micronucleus assay) (Figure 1). In parallel, additional experiments on ALI cultures were included to further characterize the models, and experiments with traditional submerged cultures were performed for comparisons. For each exposure condition, 1–2 culture inserts were included from both mono- and cocultures, and at least 3 independent experiments were performed, in order to allow for appropriate biological variation to be included in the results and analysis.

### 2.2. Nanomaterials

The Ag NM NM-300K is listed on the representative manufactured NMs list of the European Commission Joint Research Centre (JRC, Brussels, Belgium) and was selected for this study based on its toxicity in our previous work [36,37,38,39]. NM-300K was provided by the Fraunhofer Institute for Molecular Biology and Applied Ecology (Schmallenberg, Germany). NM-300K is a silver colloidal dispersion with a nominal silver content of 10% *w*/*w*. The NMs were dispersed in an aqueous solution with stabilizing agents, consisting of 4% *w*/*w* each of polyoxyethylene glycerol trioleate and polyoxyethylene (20) sorbitan mono-laurate (Tween 20). The pristine diameter of NM-300K is about 15 nm, and the size distribution is narrow, where >99% of particles (by number) have a size below 20 nm. A second peak of smaller NMs of about 5 nm has also been reported. The majority of the NMs have a spherical shape [40].

Dispersed NMs were received in vials of approximately 2.0 g each, sealed under argon. The vials were stored at room temperature (RT) in the dark before use. The dispersion medium, NM-300K DIS, contained the aqueous solution with stabilizing agents at the same concentrations as NM-300K, but without Ag. This was used as a solvent control.

### 2.3. Nanomaterial Dispersion and Characterization

Stock dispersions of NM-300K were prepared in accordance with the Nanogenotox protocol [41]. The original vial of NM-300K was vortexed (>10 s), before approximately 1 g was added to a scintillation vial (Wheaton Industries, Millville, NJ, USA). To this, water with 0.05% bovine serum albumin (BSA) was added to yield a final nominal concentration of 10 mg/mL Ag-NMs, in order to obtain a high enough concentration of Ag-NMs in the ALI exposure system. The total Ag and dissolved Ag species (<3 kDa fraction) were measured from the same samples in Camassa and Elje et al. (2022), revealing a silver concentration of 7.2 ± 0.9 mg/mL with 3.6 ± 0.1% dissolved silver species [39].

The dispersion was sonicated in an ice bath using a calibrated Q500 sonicator with a 6 mm microtip probe (Qsonica L.L.C, Newtown, CT, USA), with amplitudes of 30–40% for 7–13 min. The energy output of the sample was 1030–1285 J/mL dispersion (*n* = 10), similarly to what is recommended by the Nanogenotox protocol (1176 J/mL [41]). Additional stock dispersions were sonicated using lower energy (95–720 J/mL, *n* = 5), and were included in the study as similar results were seen compared with the other dispersions. The NM stock dispersions were kept on ice for 10 min before use, to let the NMs settle. Before use, the vial was vortexed for approximately 10 s. The dispersion was kept on ice throughout the experiment. The dispersion medium NM-300K DIS (without Ag) was prepared following the same protocol as that for NM-300K.

The NMs were previously tested for endotoxins, with endotoxin contents below the limit of detection [39]. Stock dispersions for use in the submerged exposure experiments were diluted to 2.56 mg/mL in BSA-water before further dilution in culture medium (Section 2.4) in order to ensure consistency with other studies on the same NM.

### 2.4. Physicochemical Characterization of Nanomaterials in Dispersion

NM-300K was subjected to measurement of hydrodynamic diameter and zeta potential in a Zetasizer Ultra Red (Malvern Panalytical Ltd., Malvern, United Kingdom) immediately after preparation and after 24 h. The hydrodynamic diameter was determined using dynamic light scattering (DLS) by the particles in suspension. The measured particle size is the diameter of a sphere that diffuses at the same speed as the particle being measured, which is determined by measuring the Brownian motion of the particles by DLS and then interpreting the size using the Stokes–Einstein equation.

The NM stock dispersion was vortexed and diluted 1:100 in sterile filtered MilliQ water, and a 1 mL dispersion was transferred to a disposable cuvette (DTS0012) for size analysis. The hydrodynamic diameter was measured by non-invasive back scatter at 174.7° with 3–5 steps. Analysis was performed at 25 °C with 120 s equilibration time, automatic attenuation, and no pause between steps. Data were processed in the ZS Explorer software (version 2.0.0.98, Malvern Panalytical Ltd.), using general purpose model, refractive index 1.59 and absorption 0.01.

Measurement of size distribution of NMs diluted in culture medium was performed directly after preparation and after 24 h incubation at 37 °C, 5% CO_2_. First, the stock dispersion was vortexed, and mixed with serum-free LHC-9 medium (article no. 12680013, ThermoFisher Scientific, Waltham, MA, USA) to give the highest tested concentration (141 µg/mL or 100 µg/cm^2^ in submerged exposure). Then, the sample was diluted 1:10 in sterile filtered MilliQ water, transferred to a disposable cuvette and measured as described above.

Results were presented as Z-average (Z-ave), which is the intensity-weighted mean hydrodynamic size of the ensemble collection of particles, the polydispersity index (PDI), and hydrodynamic diameter (by intensity) of individual peaks in the size distributions.

For zeta potential analysis, the NM stock dispersion was vortexed and diluted 1:100 in sterile filtered MilliQ water, and 1 mL dispersion was transferred to a pre-wetted disposable folded capillary cell (DTS1070). The zeta potential was measured at 25 °C using mixed-mode measurement phase analysis light scattering (M3-PALS).

### 2.5. Cell Culture

BEAS-2B cells, an Ad12-SV40 hybrid virus-transformed human bronchial epithelial cell line [42,43], were purchased from ATCC (Manassas, VA, USA) (SV40 immortalized, CRL-9609, LN: 62853911). The cells were cultured in serum-free LHC-9 medium without supplements, and they were maintained in an incubator with humidified atmosphere at 5% CO2 and at 37 °C. The cells were passaged two times a week at 80–85% confluency. To facilitate detachment, the cells were incubated with trypsin-EDTA (0.25%, Sigma-Aldrich, Saint-Louis, MO, USA) with polyvinylpyrrolidone (PVP, 0.5% wt/vol) for 3–5 min. Medium was added, and the suspension was centrifuged to remove the trypsin/PVP before cells were seeded at 1.3 × 10^4^ cells/cm^2^ in Corning CellBind^®^ cell culture flasks (Corning, Corning, NY, USA). The cells were used at passages (P) 3–14 (details in Appendix A).

The human alveolar type II lung epithelial A549 cells [44] were provided by ATCC, and they were cultured in Dulbecco’s Modified Eagle’s Medium, DMEM, with low glucose (D6046, Sigma-Aldrich) supplemented with 9% *v*/*v* fetal bovine serum, FBS (prod.no. 26140079, ThermoFisher Scientific), and 1% *v*/*v* penicillin–streptomycin (100 U/mL pen and 100 µg/mL strep) (catalog no. 15070063, ThermoFisher Scientific). Human endothelial EA.hy926 cells [45] were provided from ATCC and were cultured in DMEM with high glucose (catalog no. 11960, ThermoFisher Scientific), supplemented with 9% *v*/*v* FBS, 1% *v*/*v* penicillin-streptomycin, sodium pyruvate (1 mM) and L glutamine (4 mM). The cells were maintained in an incubator with a humidified atmosphere at 5% CO_2_ and at 37 °C. The cell lines were passaged two or three times a week at 85–90% confluency, using phosphate-buffered saline (PBS, catalog no. 14190094, ThermoFisher Scientific) for washing and semi-dry trypsinization using trypsin-EDTA (0.25%) incubation at 37 °C for 3 min. The cells were seeded at 1.3 × 10^4^ cells/cm^2^ in standard cell culture flasks. A549 cells were used at P2–15 and EA.hy926 cells were used at P3–19 (details in Appendix A). All cell lines were regularly tested for mycoplasma contamination and found negative.

### 2.6. Cell Cultures at the Air–Liquid Interface

The seeding of mono- and cocultures were performed in a similar manner as previously described [46,47] with some modifications. All cell types, epithelial A549 (P3–15) and BEAS-2B (P3–14), and endothelial EA.hy926 (P3–16) (details on *p* numbers in Appendix A), were seeded at a density of 1.1 × 10^5^/cm^2^. Mono- and cocultures were cultivated on permeable cell culture inserts in 6-well plates with a porous membrane of polyethylene terephthalate (PET) with a 1 µm pore diameter. Two insert types were used, with similar properties and cell attachment results: Millicell (catalog no. MCRP06H48, Sigma-Aldrich) or ThinCert™ (catalog no. 392-0128, Greiner Bio-One, Kremsmünster, Austria). The same insert type was used for all samples within an experiment.

First, the basolateral side of the membrane was pre-wetted (dipped in media), and the insert was placed upside down in the lid of a Falcon 6-well plate for inserts (catalog no. 353502, Corning). Then, 250 µL of EA.hy926 cell suspension was added to the basolateral side to reach a cell density of 1.1 × 10^5^/cm^2^. The lid with inserts was gently tilted to all sides to ensure even distribution of the cell suspension to the whole membrane surface before incubation for 3.5 h at 37 °C, 5% CO_2_. After incubation, the plate was turned in a quick movement back to the original position, and 3 mL media (for EA.hy926 cells) was added to the basolateral side. To the apical side, 1 mL of A549 or BEAS-2B cell suspension (in their own media) was added to reach a cell density of 1.1 × 10^5^/cm^2^. Monocultures of BEAS-2B or A549 were prepared in the same way, where the basolateral compartment was filled with 3 mL of media without cells. The medium volumes were optimized in pilot experiments in order to avoid too great a pressure on the cells and the insert. The cultures were incubated for 2–3 days (48–72 h) to let the cells grow to confluency. Two days’ incubation was performed only for A549 mono- and cocultures for alamarBlue and comet assay.

Epithelial and endothelial cells were seeded in their respective media. After 2–3 days of incubation of mono- and cocultures, the basolateral medium was replaced by 1.5 mL of fresh media, and the apical media was removed to place the cells in ALI conditions. For BEAS-2B/EA.hy926 a 1:1 mixture of LHC-9 and DMEM high glucose with supplements was used in the basolateral compartment, and for A549/EA.hy926 a 1:4 mixture of DMEM low glucose and DMEM high glucose with supplements was used (Appendix A). The mono- and cocultures were incubated for 20–24 h in order to let the cells adapt to ALI conditions before exposure (Section 2.7).

### 2.7. Exposure of ALI Cultures in the VITROCELL^®^ Cloud System

The VITROCELL^®^ Cloud system (6-well format) (VITROCELL® Systems GMBH, Waldkirch, Germany), was used for the aerosol exposure of mono- and cocultures at ALI conditions to NM-300K and controls. A small volume of NM dispersion or control solution was added to the Aeroneb Pro^®^ vibrating membrane nebulizer, which generates a dense cloud of droplets with a median aerodynamic diameter of 4–6 µm inside an exposure chamber. After some minutes, the humid aerosol will deposit at the bottom of the exposure chamber (area 145 cm^2^) with cell inserts [48].

Aerosol exposure was performed by aerosolizing 2 × 150 µL of sample, followed by 150 µL PBS (details below), to the mono- and cocultures positioned at ALI in the VITROCELL^®^ Cloud system at 37 °C. After 8 min, the aerosol cloud had settled, and the chamber was opened to transfer the cell inserts to 6-well plates (Falcon) with 1.5 mL fresh culture media (for mono- or cocultures). The exposure of ALI cultures was performed in the same sample order for all experiments: PBS (2 × 150 µL), NM-300K dispersion medium (2 × 150 µL), NM-300K low concentration (2 × 150 µL of stock dispersion diluted 10× in PBS or NM-300K dispersion medium), and NM-300K high concentration (2 × 150 µL of stock dispersion). In order to reduce the amount of NMs left in the nebulizer, all samples were immediately exposed to additional 150 µL PBS. Thus, all samples were exposed to a cloud with a total volume of 450 µL. All solutions and dispersions were vortexed directly before use. The nebulizer was rinsed with PBS between all exposures, and the cloud system and chamber were wiped with a tissue with ethanol.

The relative amount of nebulized solution that is deposited on top of the cells, the deposition efficiency, can be measured by comparing the amount of deposited substance on the insert to the original solution, either by using a fluorescent compound or elemental analysis. As the deposition efficiency can vary between different nebulizers, the same nebulizer was used for all experiments in this study. We previously measured the deposition efficiency of this nebulizer to be 53% [39]. Additionally, the deposition of Ag in NM-300K was measured giving similar results [39]. This information was used to choose the exposure volumes needed for achieving the intended nominal concentrations for cell exposure.

### 2.8. Positive Control Exposures

Exposure of ALI cultures to positive controls were performed via the basolateral culture media below the inserts in 6-well plates, for 20–24 h. First, stock solutions were prepared and stored for use within all experiments before being diluted in sterile filtered H_2_O directly before use and further diluted in culture medium. Chlorpromazine hydrochloride (catalog no. C8138, Sigma-Aldrich) was used as a positive control for cytotoxicity in the alamarBlue assay, with a stock solution at 5 mM in H_2_O stored at 4 °C and exposure concentration of 50–100 µM. Mitomycin-C (catalog no. A2190.0002, PanReac AppliChem [VWR/Avantor]) was used as a positive control for micronuclei induction in the micronucleus assay, with stock solution at 0.2 mg/mL in dimethyl sulfoxide (DMSO) stored at −20 °C, and exposure concentration at 0.15 µg/mL, similarly to Reference [49]. In one experiment with A549/EA.hy926 and BEAS-2B/EA.hy926 cocultures, a higher concentration (0.30 µg/mL) was additionally included.

### 2.9. Characterization of the ALI Cultures by Microscopy Analysis

Daily evaluation of cell density and proliferation was performed with a Leica DM-IL microscope. More detailed characterization was performed by confocal microscopy. For confocal microscopy, cells in separate culture inserts were stained, fixed, and mounted between two glass coverslips, as described in Reference [39]. In brief, the plasma membranes were stained with CellMask Deep Red Plasma Membrane Stain (Invitrogen, 1:750 dilution in serum-free medium, 15 min at 37 °C), and the cells were fixed in formaldehyde (4%, 15 min, RT), before the nuclei were counterstained with DAPI (4′,6-diamidino-2-phenylindole, ProLong Gold Antifade reagent with DAPI, ThermoFischer Scientific). Confocal microscopy was performed using a Zeiss LSM 700 (lasers 405 and 639 nm; objective 40×). Image acquisition and processing were performed with the Zeiss Software ZEN. Z-stack acquisition was performed with 12–52 μm thickness with 7–53 images for each stack.

### 2.10. Barrier Function of the ALI Cultures by Elemental Analysis and Fluorescence

The barrier function of the BEAS-2B monocultures and BEAS-2B/EA.hy926 cocultures at the ALI, simulating the human lung–blood barrier, was tested by measuring the permeation of Ag or a fluorescent hydrophilic molecule into the basolateral medium. The procedures were performed similarly to as described in Reference [39] for A549 and A549/EA.hy926 cultures.

The permeation of Ag through the barrier was measured by analyzing the total Ag in the basolateral medium of the ALI cultures after 20–24 h exposure, and it was compared with the deposited Ag on empty culture inserts. The basolateral medium was collected in Eppendorf tubes, stored at −20 °C, and used for cytokine analysis by ELISA (Section 2.11) and Ag analysis by inductively coupled plasma mass spectrometry (ICP-MS). Culture medium was thawed on ice, vortexed for 10 s and 250–500 µL was transferred to a Teflon container (18 mL) vial. Sample preparation and ICP-MS analysis was performed as described in our recent study [39]. The ICP-MS results were analyzed to give the total Ag mass per insert, which was then divided by the deposited mass (low: 0.8 µg/cm^2^, high: 6.0 µg/cm^2^ [39], multiplied by insert area), and multiplied by 100% to give the percentage of Ag permeation through the ALI cultures.

Breakthrough of fluorescein sodium salt was measured on separate cell cultures in order to avoid interference between fluorescein and alamarBlue solution. After exposure to PBS (Section 2.7), 150 µL of fluorescein sodium salt (10 µg/mL in PBS) was nebulized and deposited on top of the cells for 3.5 min. In parallel, fluorescein was deposited on empty inserts in order to estimate the maximum leakage through the insert without cells and on inserts filled with 1 mL PBS in order to measure the maximum deposited fluorescein in the apical side. The leakage samples were transferred to 6-well plates with 1.5 mL medium on the basolateral side, and samples for deposition efficiency were transferred to empty wells. After 22–24 h of incubation, the fluorescence of fluorescein was measured in the basolateral medium or in apical PBS, related to a seven-point fluorescein standard curve (1.6–50 ng/mL) and blank in the respective medium or PBS. Fluorescence was read in triplicate (90 µL/well) in a black 96-well plate on a FLUOstar OPTIMA microplate reader (BMG Labtech, Ortenberg, Germany) with excitation 480 nm and emission 525 nm. Two independent experiments (*n* = 2) were performed, each with 1–2 culture inserts for deposition and breakthrough.

### 2.11. Cytokine Measurement

The basolateral medium from the exposed mono- and cocultures were transferred to Eppendorf tubes and frozen at −20 °C. The amounts of cytokines present in the basolateral media was measured by the enzyme-linked immunosorbent assay (ELISA), which is a commonly used colorimetric immunological assay. The target molecule in the sample will bind to a specific antibody immobilized at the bottom of the microplate well. Through the addition of the second antibody, a sandwich complex is formed. A substrate solution binds to this complex and produces a measurable signal, which is directly proportional to the concentration of target present in the original sample.

ELISA was performed using kits for the human cytokines IL-6 (prod.no. 88-7066, Invitrogen) and IL-8 (prod.no. 88-8086, Invitrogen). The manufacturer’s recommended procedures were followed. The samples were thawed on ice and vortexed before use and diluted 1–200 times in the assay buffer to fit within the measurement region. A standard curve was included in all plates. Duplicate measurements from each sample were performed. Plate washing was performed on a Hydroflex (TECAN, Grödig, Austria) microplate washer. Absorbance was read at 450 nm on an Infinite 200 Pro M Nano (TECAN) plate reader.

Potential interference between NM-300K and the performance of the ELISA was investigated. NM-300K was prepared as described above and diluted in cell culture media for BEAS-2B cells, A549 cells, or EA.hy926 cells, in order to achieve concentrations of 30, 3 and 0.3 µg/mL. The NMs in media were added to the ELISA plate in duplicates and mixed with reagent buffer or standard (final concentrations 25 pg/mL IL-6 or 31 pg/mL IL-8) provided in the kit. Further steps in the assay were run as described for the other samples.

### 2.12. Cell Viability Assessed by alamarBlue Assay

Cell viability was determined by the alamarBlue assay, which is based on the metabolic activity of cells and is commonly used for the quantitative analysis of cell viability and proliferation. The active ingredient in alamarBlue reagent (Sigma-Aldrich) is resazurin, which is a blue non-toxic, cell-permeable compound with low fluorescence. In living cells, resazurin is reduced to resorufin which is red and highly fluorescent, and the color change is detected on a plate reader.

AlamarBlue assay was performed 20–24 h after cell exposure. First, the basolateral media of ALI cultures was removed and saved for cytokine analysis. For monocultures, 1 mL alamarBlue reagent 10% *v*/*v* in cell culture media was added to the apical compartment, and 1.5 mL alamarBlue-free media was added to the basolateral compartment. For ALI cocultures, coculture media with alamarBlue 10% *v*/*v* was used in both compartments with the same volumes as for the monocultures. The plates were incubated for approximately 1 h. The plates were gently swirled to ensure even distribution of the alamarBlue solution, and 40 µL aliquots were transferred in triplicate to black 96-well plates, before fluorescence (excitation 530 nm, emission 590 nm) was measured on a FLUOstar OPTIMA microplate reader. Blank values (alamarBlue medium without cells present) were subtracted from the fluorescence intensity, which was further normalized by the average measurement of negative control (incubator control) set to 100% relative viability. Potential interference of NM-300K with the alamarBlue assay was investigated as described in Appendix A.

### 2.13. Cell Detachment and Counting

Directly after performing the alamarBlue assay (Section 2.11), both sides of the insert were washed with PBS. Cells were detached by trypsin-EDTA incubation and subsequent mixing/washing of insert. For the apical compartment with BEAS-2B or A549 cells, 300 µL trypsin-EDTA was used (with PVP for BEAS-2B). For the basolateral compartment, PBS was used for monocultures and 1–1.5 mL trypsin-EDTA 0.05% (Sigma-Aldrich) was used for EA.hy926 cells. Inserts were trypsinized for 3–5 min at 37 °C, and the apical suspension was mixed with a pipet to facilitate detachment before medium for each cell type was added (1 mL in apical side, 3 mL in basolateral side). BEAS-2B cells were centrifuged at 200 g for 5 min and resuspended in 1 mL fresh culture medium to remove trypsin-EDTA/PVP which was not neutralized by the serum-free cell culture medium.

The cell suspension was mixed 1:1 with trypan blue (0.4%, Invitrogen) for the staining of cells with compromised cell membrane. The cells were counted in an automated cell counter (Countess^®^ C10227, Invitrogen) in order to determine the total number of live cells and viability (%). The cell density was calculated by dividing the total number of live cells by the membrane insert area. Immediately after counting, the cell suspensions were further diluted to approximately 200,000 live cells/mL and used for genotoxicity studies (Section 2.14 and Section 2.15).

### 2.14. DNA Damage Assessed by the Comet Assay

Cell suspensions from ALI and submerged cultures (Section 2.16) were subjected to DNA damage evaluation by the enzyme-modified version of the comet assay. Briefly, in the comet assay, cells are embedded in gels, lysed, and the remaining nucleoids are subjected to an electrophoretic field. The movement of damaged DNA causes comet formations, wherein the relative amount of DNA in the comet tail is proportional to the number of DNASBs.

Reagents used for the comet assay were provided by Sigma-Aldrich unless otherwise stated. A cell suspension with approximately 10,000 cells in 50 µL was mixed with 200 µL low melting point (LMP) agarose (0.8% in PBS) in a 96-well plate, yielding a final concentration 0.64% LMP agarose. Mini-gels (10 µL) were placed on coded microscopy slides precoated with 0.5% standard melting point agarose, and the slides were submerged in lysis solution (2.5 M NaCl, 0.1 M EDTA, 10 mM Tris, 1% *v*/*v* Triton X-100, pH 10, 4 °C) overnight.

For the detection of oxidized bases, the bacterial repair enzyme formamidopyrimidine DNA glycosylase (Fpg, gift from NorGenoTech, Oslo, Norway), which converts oxidized bases to SBs, was used [11]. After lysis, gels for Fpg treatment were washed twice for 8 min in buffer F (40 mM HEPES, 0.1 M KCl, 0.5 mM EDTA, 0.2 mg/mL BSA, pH 8, 4 °C), before being placed in a humid box and covered with Fpg (200 µL/slide) and polyethylene foil. Fpg incubation was performed for 30 min at 37 °C.

All slides were placed in the tank and submerged with electrophoresis solution (0.3 M NaOH, 1 mM EDTA, pH > 13, 4 °C), and DNA was allowed to unwind for 20 min. The electrophoresis was run for 20 min (25 V, 1.25 V/cm, Consort EV202). Gels were neutralized in PBS, washed in ultrapure H_2_O, and air-dried overnight. Staining of DNA was performed with SYBR gold (1:2000), and scored in Leica DMI 6000 B (Leica Microsystems), equipped with a SYBR^®^ photographic filter (ThermoFischer Scientific) using the software Comet assay IV 4.3.1 (Perceptive Instruments, Bury St Edmunds, UK). Comets were scored semi-blindly by two operators, where all slides within one experiment were scored by the same operator. Median DNA tail intensity was calculated from 50 comets per gel as a measure of DNA SBs. Medians were averaged from 2–6 gels per sample per *n* = 3–7 independent experiments.

Hydrogen peroxide (H_2_O_2_) was used as a positive control for DNA SBs. Cells from negative control inserts were embedded in gels and submerged in 13–100 µM H_2_O_2_ in PBS for 5 min at 4 °C. The samples were washed twice for 2 min in PBS (4 °C) and then submerged in a separate Coplin jar of lysis solution. The short time between H_2_O_2_ treatment and lysis limits the process of damage repair. The H_2_O_2_ exposure experiments with BEAS-2B ALI mono- and cocultures were conducted by placing all cell types (BEAS-2B from monoculture, BEAS-2B from coculture, and EA.hy926 cells from coculture) on the same slide in order to minimize variation. For experiments with submerged cultures (Section 2.16), all cell types and exposure conditions were placed on the same slide. As a negative control for the H_2_O_2_ exposure experiments, a separate slide with gels was exposed to PBS in parallel with H_2_O_2_ exposure.

As a positive control for the function of the Fpg enzyme, A549 cells were exposed to a photosensitizer, Ro 19–8022 (Hoffmann La Roche, Switzerland), and irradiated with visible light before embedding in gels, as described in Elje et al. 2019 [50]. The photosensitizer Ro 19-8022 induces with light oxidized purines, mainly 8-oxoG, which is detected by the Fpg [11]. The function of Fpg was controlled on a regular basis and was not included in all experiments. The positive control had an expected effect compared to the historical control data, with a net Fpg (level of SBs + Fpg minus level of SBs) of >20% DNA in tail.

### 2.15. Chromosomal Damage Assessed by the Cytokinesis-Block Micronucleus Assay

In parallel to the comet assay, cells from ALI cultures were seeded for detection of chromosomal damage by the cytokinesis-block micronucleus assay. The micronucleus assay measures the ability of the test substance to induce structural chromosome damage (clastogenic effect) or numerical chromosome alterations (aneugenic effect). Micronuclei are formed from chromosome or chromatid fragments or from whole chromosomes that lag behind in cell division. The addition of the active polymerization inhibitor cytochalasin B allows for analysis of the micronuclei frequency in cells that have completed one mitosis after treatment with the test substance, as such cells which are binucleated because the cytochalasin B prevents the separation of daughter cells after mitosis [13,51].

After the treatment and detachment of cells from ALI cultures, approximately 0.5–1 × 10^5^ cells were seeded on flame-sterilized coverslips placed in 6-well plates with 1.5 mL media to a final concentration of 6 µg/mL cytochalasin B (prod.no. C6762, Sigma-Aldrich). The coverslips were incubated with culture media for 1–3 h before adding cells in order to facilitate cell adhesion. Coverslips and plates used for BEAS-2B cells were first coated with collagen IV (prod.no. 804592, Sigma-Aldrich). To each well with a coverslip, 1 mL collagen IV (30 µg/mL) diluted in Hanks’ Balanced Salt solution (HBSS) (prod.no. 14175046, ThermoFisher Scientific) was added, before overnight incubation at 4 °C. The coated plates were washed with PBS before medium with cells was added. Cells were incubated at 37 °C and 5% CO_2_ for 26–33 h (1–2 cell cycles). The same incubation time was used for all samples within the same experiment.

Cells were washed with PBS and fixed with methanol and acetic acid (3:1) in two steps, first for 15 min at RT, and then for up to 4 days at 4 °C. Coverslips were dried in the fume hood for 3–5 min and mounted on coded standard microscopy slides cleaned with ethanol, with a drop of the mounting medium ProLong Gold Antifade reagent with DAPI (ThermoFischer Scientific).

Cells were imaged in Zeiss Imager-Z2 microscope with a Metafer camera (MetaSystems Hard & Software GmbH, Altlussheim, Germany) and analysis system for micronuclei scoring. Scoring was performed in a semi-automatic manner, using 10× and 40× objectives. More details on the system and settings used for scoring can be seen in Appendix A, including the percentages of analyzed binucleated cells (Appendix A). The selected settings for scoring and analysis did not identify all binucleated cells with micronuclei, giving some false negative and false positive cells. Thus, to avoid this, all identified binucleated cells were manually accepted or rejected, and cells with possible micronuclei were checked with a 40× objective.

### 2.16. Statistical Analysis

At least three independent experiments were performed for each test method, unless otherwise stated. In each experiment, 1–2 parallel culture inserts were included (Appendix A). Results are presented as mean with standard deviation (SD) calculated from the average results from *n* experiments. Normal distribution of data was assumed. In order to evaluate the statistical significance of the results, one-way ANOVA was performed followed by Dunnett’s multiple comparisons test using GraphPad Prism version 9.3.1 for Windows (GraphPad Software, San Diego, CA, USA). Results were compared to the PBS control for ALI cultures and unexposed control for submerged cells. Fluorescein permeation results were compared to empty inserts with no cells. Ag permeation results were analyzed by one-way ANOVA with post-test Sidak to allow for multiple comparisons between low and high concentrations and between models and empty inserts (16 comparisons in total). The level of significance was set to *p* < 0.05. Calculation of EC_50_ values, the concentration giving 50% response, was performed using non-linear regression analysis with the Hill equation, in GraphPad Prism. Mathematical calculations were performed in Excel (Microsoft 365).

## 3. Results

### 3.1. Nanomaterial Dispersion Quality and Physicochemical Characterization

The hydrodynamic diameter of NM-300K in stock dispersion and diluted in LHC-9 culture medium was measured by DLS directly after preparation and after 24 h (Table 1). The NMs in stock dispersion had a hydrodynamic diameter (Z-ave) of 149 nm and were stable in dispersion for 24 h. The size distribution of the NMs had two or three peaks, where most particles were within the peak with mean size 110–265 nm, and with some smaller (5–40 nm) and larger (>3000 nm) particles. The zeta potential was −17.1 ± 2.8 mV, indicating that the NM dispersions were semi-stable.

NM-300K diluted in LHC-9 culture medium had a higher hydrodynamic diameter than the stock dispersion, with a Z-ave of 378 nm and main peak with slightly higher size as for stock dispersion. After 24 h, larger NMs were detected in some of the measurements, giving a Z-ave between 119–5180 nm and a high PDI. Medium without NM-300K had a Z-ave of 15.4 ± 1.0 nm with PDI 0.349 ± 0.054 (*n* = 2).

### 3.2. Characterization of the Advanced Models

The advanced models of BEAS-2B/EA.hy926 and A549/EA.hy926 cells, cultured at the ALI, were characterized for cell density, viability and barrier integrity. The ALI-cultured apical cells were moist with a shiny surface, though some cultures occasionally showed a drier appearance (Appendix A). BEAS-2B cells were grown in dense structures on the porous membranes, as seen by confocal microscopy (Figure 2A,B). Higher density was observed for BEAS-2B in coculture with EA.hy926 cells compared to monocultures. Some holes in the apical cell layers of both mono- and cocultures were observed, and the fewest holes were seen in A549 cocultures (details not shown). The confocal images of BEAS-2B cells also indicated slightly higher thickness of the apical cell layer in cocultures, with cells growing in multilayers, compared to the BEAS-2B cells in monocultures (Figure 2 and Appendix A). Cell counting after detachment of cells confirmed a higher density of BEAS-2B cells in coculture compared to BEAS-2B in monocultures, indicating a higher proliferation of the cells in this condition (Table 2). The opposite result was seen for A549 cells, where the density was higher in monoculture compared to coculture with EA.hy926 cells. The endothelial cells had a similar density of both types of cocultures (Table 2) and were growing in a confluent monolayer (Figure 2C). The collected cells had high viability, though some cells were lost during the detachment process and during washing before trypsinization, and some cells still remained on the insert.

The barrier integrity of the advanced models was investigated by measuring the permeation of the water-soluble fluorescein sodium salt and Ag (NMs or dissolved species) from NM-300K after aerosol exposure through the cellular layer by quantification in the basolateral media after 24 h (Table 3). A high permeation of fluorescein, at the same level as empty inserts without cells, was found in BEAS-2B/EA.hy926 cocultures (70%) (Table 3). Strongly reduced fluorescein permeation was seen in BEAS-2B monocultures (20%) and in A549/EA.hy926 cocultures (9%). No difference was seen in the permeation of fluorescein between incubator control and PBS-exposed cultures.

The permeation of Ag was higher in BEAS-2B monocultures compared with cocultures (Table 3). This difference was highest after exposure to the low concentration of NM-300K, giving a permeation of 57% in monocultures and 37% in cocultures. In the basolateral medium of cultures exposed to the high concentration of NM-300K, the Ag concentrations of both mono- and cocultures were similar to the maximum permeation through empty inserts (11 µM) (no statistically significant difference by one-way ANOVA with post-test Sidak, *p* > 0.05). The barrier integrity of A549 cultures differed from the BEAS-2B cultures. A slightly higher Ag permeation was seen after exposure to low concentration of NM-300K in A549 cocultures (14%) compared to monocultures (8%) although the difference was not statistically significant. However, the permeability of both A549 mono- and cocultures was lower than for BEAS-2B mono- and cocultures (*p* < 0.05). After exposure to the high concentration of NM-300K, similar results were seen for both A549 and BEAS-2B models, where the permeability was about the same as for the maximum permeation through empty inserts (*p* > 0.05). A low concentration of Ag was found in the basolateral medium of cultures exposed to NM-300K DIS, and the concentration was similar for all models.

### 3.3. Toxic Responses after NM-300K Exposure in Advanced Respiratory BEAS-2B or A549 Models

#### 3.3.1. Cytotoxicity

The mono- and cocultures of BEAS-2B/EA.hy926 or A549/EA.hy926 cells were exposed to aerosolized NM-300K and control solutions in the VITROCELL^®^ Cloud. After 20–24 h, cytotoxicity was investigated by the alamarBlue assay following the experimental design in Figure 1. The measured deposited concentrations of NM-300K (low and high), with nominal concentrations 1 and 10 µg/cm^2^, were measured in our recent study to be 0.8 and 6.0 µg/cm^2^, respectively [39].

Cell viability is presented relative to incubator control (NC, set to 100%) and statistically analyzed against the PBS control. A reduction in the relative cell viability was seen after NM-300K exposure at high concentration in BEAS-2B monocultures (57%, Figure 3A), but not in the cocultures compared with the PBS exposure control. The viability of BEAS-2B cells in cocultures was significantly reduced after aerosol exposure to PBS, with a relative viability of 67%, compared with the incubator control. This effect of PBS was not seen in the monocultures (Figure 3B). The viability of cells exposed to NM-300K DIS was similar to that of cells exposed to PBS, in both models. The viability of EA.hy926 cells was not affected by aerosol exposure to NM-300K or PBS. The positive control, 50–100 µM chlorpromazine hydrochloride in basolateral media, strongly reduced the viability in all cultures, as expected.

Similar results as with the BEAS-2B models were seen with A549 monocultures and A549/EA.hy926 cocultures after NM-300K exposure (Table 4, details in [39]). The viability of EA.hy926 cells was reduced after aerosol exposure when in coculture with A549 but not with BEAS-2B (Table 4).

For the comparison of the new advanced models with the corresponding traditional cell models, the cytotoxicity of NM-300K was also tested with submerged exposure of monocultured cells by alamarBlue assay. NM-300K was cytotoxic in BEAS-2B cells at concentrations above 10 µg/cm^2^ and at 10 µg/cm^2^ for submerged and ALI exposure, respectively (Figure 3 and Appendix A). BEAS-2B cells were more sensitive to NM-300K exposure compared to A549 and EA.hy926 cells in submerged conditions (Appendix A, [52,53]). No interference with the alamarBlue assay was detected for NM-300K (Appendix A).

#### 3.3.2. Secretion of Pro-Inflammatory Cytokines IL-6 and IL-8

Upon NM exposure, pro-inflammatory cytokines can be secreted by the airway epithelium and endothelium in order activate the immune system. The concentrations of the pro-inflammatory cytokines IL-6 and IL-8 secreted from the ALI cultures into the basolateral medium during exposure were measured by ELISA. The results are presented as absolute concentrations (Figure 4 and Appendix A) and relative to NC (Appendix A).

An apparent trend towards increased levels of IL-8 was seen after NM-300K exposure in all cell models; however, a statistically significant increase was measured only for BEAS-2B in mono- and coculture for the lowest concentration of NM-300K compared with untreated incubator control. A similar effect was seen on IL-6 levels in BEAS-2B cocultures only. In BEAS-2B mono- and cocultures, the concentrations of both IL-6 and IL-8 were higher for low-concentration NM-300K compared with the high concentration. There was a significant effect of PBS exposure on the levels of IL-6 in monocultures of BEAS-2B compared to NC (Figure 4 and Appendix A).

IL-6 and IL-8 concentrations were found to be higher (3× and 9×, respectively) in cocultures of BEAS-2B/EA.hy926 compared with BEAS-2B monocultures (Figure 4 and Appendix A). For the monocultures, the increase in IL-6 was about the same for low-concentration NM-300K and NM-300K DIS. However, no increase was detected after exposure to the high concentration of NM-300K, for which the concentrations of NM-300K DIS was matching. For the A549 models, the level of IL-8 was about 4× higher in monocultures than in cocultures. For IL-6 level, there was no difference between mono- and cocultures of A549 cells. No interference between the NM-300K and the assay was found (Appendix A).

#### 3.3.3. Genotoxicity by DNA and Chromosomal Damage in ALI Cultures

After viability analysis by alamarBlue assay and cytokine secretion analysis by ELISA, cells from the same samples were analyzed for DNA damage by the comet assay and cytokinesis-block micronucleus assay.

Different sensitivities on induction of DNA SBs and oxidized base lesions after exposure to NM-300K were measured by the enzyme-modified comet assay when comparing the different cell types and models. NM-300K exposure at low concentrations induced an increase in DNA SBs and SBs + Fpg in BEAS-2B cells in monoculture, with 26 ± 18% DNA in the tail (Figure 5A), but no effect in cocultures (Figure 5B). No significant effect was measured in BEAS-2B or in EA.hy926 cells (Figure 5) after exposure to high-concentration NM-300K. The levels of DNA SBs were similar in the incubator control (NC) and in samples exposed to PBS. No effect of NM-300K DIS was seen. However, the background of DNA SBs (in NC) was slightly higher in BEAS-2B cells from cocultures (5.9 ± 3.6% DNA in tail, *n* = 3) compared with monocultures (1.4 ± 1.6% DNA in tail, *n* = 6). For comparison, submerged NM300-K exposure of BEAS-2B cells did not induce any genotoxicity, as an increase in SBs was detected only at cytotoxic concentrations (from 10 µg/cm^2^) (Appendix A).

H_2_O_2_ exposure induced a concentration-related induction of SBs in BEAS-2B cells from monocultures and EA.hy926 cells from cocultures (Figure 6 and Table 5). BEAS-2B cells in cocultures were found to be more sensitive, with a high level of SBs also at the lowest concentrations of H_2_O_2_ (80 ± 11% DNA in tail at 13 µM H_2_O_2_). Different media compositions were used in the BEAS-2B monocultures (LHC-9) and in the BEAS-2B/EA.hy926 cocultures (DMEM and LHC-9, 1:1). In submerged cells, which showed less sensitivity to H_2_O_2_ exposure than ALI cultures, the different media compositions did not affect the viability (Appendix A) or H_2_O_2_ sensitivity (Table 5 and Appendix A). Cells from A549 mono- and cocultures had a high response to 100 µM H_2_O_2_, as expected, and only one concentration was tested.

No significant effect on micronuclei induction was found after exposure to PBS, NM-300K DIS, or NM-300K, on any of the cultures, compared to the PBS control (Figure 7). A high level of micronuclei was induced by the positive control (0.15 µg/mL mitomycin-C in basolateral media) in BEAS-2B and EA.hy926 cells from mono- and cocultures (Figure 7A,B), and slightly lower for A549 in mono- and cocultures (Figure 7C,D). The effect of mitomycin-C in A549 monocultures was significantly different from that of the unexposed NC control only (*p* = 0.04), and the increase in micronuclei formation was not statistically significant from the PBS control (*p* = 0.08). The proportion of binucleated cells in the samples for micronuclei investigation was estimated to be 21% for BEAS-2B and 19% for EA.hy926 cells. Corresponding numbers for A549/EA.hy926 cocultures were about 21% for A549 and 8% for EA.hy926 cells (Appendix A).

## 4. Discussion

An important part of NAMs, which are essential for NGRA, is the development and characterization of advanced in vitro models. Advanced respiratory in vitro models are of high importance for the hazard assessment of NMs after inhalation exposure. Cells cultured and exposed at the ALI represent a more physiological scenario than cells in submerged conditions. In order to develop the most realistic NAMs, the characterization, testing and validation of models is needed. Of importance to this is comparison of the effects of reference NMs on different advanced models for the same target, as well as to benchmark the effects of the tested NMs against the effects in traditional 2D in vitro models. This study focused on the characterization and application of the immortalized human bronchial epithelial cell line BEAS-2B cultivated at ALI in monoculture or in cocultures with endothelial EA.hy926 cells for the testing of different toxicity endpoints. It is, to our knowledge, the first study to successfully apply several genotoxicity endpoints, including the micronucleus assay, in advanced BEAS-2B cocultures at the ALI, and to perform a comparison of the effects of a reference NM in mono- and cocultures, and also with the more extensively used human alveolar epithelial cell line A549. Further, effects on the advanced models at the ALI were compared with responses in traditional corresponding submerged cultures in monocultures.

Cell growth, barrier integrity, and confluency differed between mono- and cocultures of BEAS-2B cells. In cocultures with EA.hy926, the BEAS-2B cells had lower confluency compared with monocultures, and they showed a multilayer growth, which was not seen with the A549 cells. Thus, more holes were seen under confocal microscopy in the BEAS-2B epithelial layer, which was also thicker. We found a higher density of BEAS-2B cells in cocultures compared with monocultures, indicating higher cellular growth and stimulated cell proliferation in the cocultures. The opposite was found with A549 cells, where the cells had lower density in cocultures compared to monocultures. The stimulated proliferation of BEAS-2B cells in cocultures with EA.hy926 cells was found not to be due to different media compositions of mono- and cocultures. Rather, the increased proliferation of BEAS-2B cells in coculture might be related to cell signaling from the endothelial cells. The multilayer growth of BEAS-2B cells in coculture indicates that the advanced model facilitates conditions similar to tissue physiology in the lungs, as has been shown also in previous studies [10,54].

In monoculture, BEAS-2B and A549 had similar cell numbers at the end of the cultivation period, despite the longer doubling time (4 h) of BEAS-2B cells compared to A549 [55,56]. The endothelial cells showed a similar density and appearance in both coculture types, with BEAS-2B and A549, respectively.

The barrier function appeared to be less in BEAS-2B cocultures than in monocultures, as indicated by the higher permeability of fluorescein, which was measured in the basolateral medium. This finding is in line with the higher frequency of observed holes in the apical cell layers of cocultures of BEAS-2B. In contrast, Ag permeation was higher in BEAS-2B monocultures than in cocultures. It is known that Ag has a high affinity for sulfur and that it may form toxic complexes with sulfur-containing proteins in the cells [57]. The lower permeation of Ag in the coculture may be explained by the additional barrier constituted by the endothelial cells and the measured higher epithelial cell density in cocultures, and thus increased interaction with or dissolution of the NMs. Ag permeation after exposure to NM-300K at high concentrations was in all culture types similar to the permeation through empty inserts. As total silver was measured, it included both particles and dissolved Ag-ions diffusing from the apical side down the concentration gradient. However, the permeation of Ag was similar after exposure to low and high concentrations, and it was not increased as may be expected by the larger gradient. This low permeability might be due to the agglomeration/aggregation of the nanoparticles at high density, making them less likely to penetrate into the pores of the insert membrane, the pores being blocked, a saturation of the medium, or a combination of these factors.

A549 mono- and cocultures showed less permeation of fluorescein and also of Ag after exposure to low-concentration NM-300K, than BEAS-2B cultures. This is in accordance with previous studies showing that A549 cells form tight junctions [39,46] and thus a stronger barrier, measured as trans-epithelial resistance (TEER), at an earlier stage than BEAS-2B cells [54]. Different bronchial cell lines, including BEAS-2B, cultivated at ALI, were evaluated for barrier functions by He et al., (2021), and only Calu-3 cells were found to sustain a strong TEER for up to 21 days. Also, the immortalized cell line 16HBE formed tight junctions and developed a strong TEER, though the TEER dropped considerably in ALI conditions [10].

In order to compare the responses of the different models to a toxic insult, the cells were exposed to Ag NM-300K, which is a reference NM commonly used in many projects related to the safety of NMs (such as the European Commission FP7 project NanoReg, and H2020 projects NanoReg2, PATROLS and RiskGONE). The relative cell viability of BEAS-2B monocultures at the ALI was reduced by NM-300K exposure at the highest concentration. This is similar to what we observed for A549 monocultures [39]. The viability was reduced at lower concentrations after ALI exposure compared with submerged exposure, which may indicate higher sensitivity of the ALI models. However, direct comparisons between the models are difficult due to large differences in the experimental conditions including aerosol or submerged exposure, cell number, and cell density.

Significant reduction in cell viability was only measured after exposure to high concentrations of NM-300K in monocultures. In cocultures, we observed that PBS aerosol exposure reduced cell viability of the BEAS-2B cells, compared with incubator control, and thus no significant difference in viability could be detected between PBS and NM exposures. The same effect of PBS was also seen previously with A549 cocultures [39]. This indicates that the coculture conditions at the ALI sensitize the apical epithelial cells to the aerosol exposure both of non-toxic and toxic compounds. One could speculate that the lack of direct basolateral contact with the cell culture media in the cocultures could influence this, due to the confluent layer of the endothelial cells. Additionally, the interaction and interplay between the endothelial cells in itself may play a role. Interestingly, the viability of the endothelial cells was not reduced after PBS exposure in coculture with BEAS-2B but with A549 [39], despite the demonstrated stronger barrier of the A549 epithelial layer.

When genotoxicity was tested, the models showed different sensitivities. Genotoxicity measured as DNA SBs was found only in BEAS-2B monocultures at low concentrations of NM-300K. The lack of genotoxic effects at high concentrations could be due to loss of damaged cells during the washing steps, as cytotoxicity was measured at this concentration. No genotoxic effect was seen in BEAS-2B or in the EA.hy926 cells in coculture. The lack of an effect in the BEAS-2B cells in coculture may be due to the higher cell density and thereby relatively lower number of particles per cell. One may also speculate as to whether the co-cultivation with the endothelial cells may increase the emergency preparedness of the BEAS-2B cells towards toxic insults. The increased level of pro-inflammatory cytokines in the cocultures (Figure 4) may support such a theory, and it has been suggested previously that increased expression of cytokines, such as IL-6, can promote DNA repair [58]. In cocultures of A549 and EA.hy926, genotoxicity measured as SBs was only found in the endothelial cells. However, we previously showed that in triculture with addition of differentiated THP-1 cells, no SBs were detected [39]. These results are not false negative, as NMs were internalized in the A549 cells [39]. A genotoxic response can be secondary due to the induction of an inflammatory response. Thus, we measured IL-6 and IL-8 levels in mono- and cocultures. No significant increase in IL-6 or IL-8 levels was detected in the A549 models after NM-300K exposure. In general, there was a trend towards increased levels after ALI exposure. However, the basal level of IL-8 was much higher in A549 cocultures than in monocultures, which was the opposite as seen with BEAS-2B. For BEAS-2B cocultures, the level of both IL-6 and IL-8 was increased after exposure to low concentration of NM-300K, although statistical significance was reached only for IL-6. The level of IL-6 was also increased in BEAS-2B monocultures at low concentrations of NM-300K, but it was about 9× lower than for cocultures. In contrast, significant induction of SBs was measured only in monocultures at low concentrations of NM-300K. As the toxic response was changed when coculturing different cell types, as compared with monocultures, further studies are merited to elucidate the interplay between the cell types and the importance of coculturing multiple cell types for hazard identification.

Genotoxicity was further tested at the chromosomal level by the micronucleus assay. Few studies have applied the micronucleus assay on advanced human lung models at the ALI; at the time of writing, we found only one publication on monocultured A549 [49], one on monocultured BEAS-2B and other bronchial cell types [59], and four studies on nasal epithelial cells from donors [60,61,62,63]. We successfully employed this assay to both mono- and cocultures of BEAS-2B cells exposed at the ALI. The effect of the positive control mitomycin-C (0.15 µg/mL in basolateral medium) was found to be more pronounced in the cocultures. The effect of mitomycin-C on A549 mono- and cocultures was lower than on BEAS-2B, and the effect in A549 monocultures was only significantly different from the unexposed control and not from the PBS control. Mitomycin-C also induced micronuclei in the underlying endothelial cells in coculture with the BEAS-2B-cells, but not with A549 cells, and the proportion of binucleated cells was lower when cocultured with A549 cells. The epithelial and endothelial cells were cultured with cytochalasin B for the same duration; however, a longer incubation time was used for BEAS-2B/EA.hy926 cocultures compared to A549/EA.hy926 cocultures due to the differences in cell doubling times. Further investigations are needed in order to determine whether the differences in response are due to lower sensitivity towards mitomycin-C when in coculture with A549 or if experimental optimization is needed to produce a higher proportion of binucleated cells.

No aneugenic or clastogenic effects were detected after NM-300K exposure at the ALI, which is in contrast to previous studies with significant micronuclei induction by NM-300K in submerged BEAS-2B cells [64,65]. ALI exposure was performed when the cells had developed a confluent cell layer, which may have affected the cell cycle and proliferation rate. In our study the cells from ALI cultures were directly seeded with cytochalasin B on coverslips after NM treatment, at a lower density to initiate DNA replication and MN formation, before the cell fixation and analysis of MN in binucleated cells. The discrepancy in the effects between submerged and ALI may, therefore, be due to their different stages in division cycles during NM exposure, which merits further experiments to optimize the method. The agglomeration state of the NMs would also influence their toxicity, and this might differ between submerged and ALI exposure. A previous study on titanium dioxide NMs in BEAS-2B cells showed that changes in the composition of exposure medium affected the induction of MN due to differential states of agglomeration [66].

The experimental design presented in this study, enabling several endpoints to be measured from each insert, allows for increased throughput, reduced costs, time and materials and thus better sustainability, compared to measuring all endpoints in separate inserts. This is an important aspect of the NAMs and crucial for an integrated approach to testing and assessment (IATA) in NGRA. Further, more direct comparisons are enabled, as the different endpoints are measured from the same exposure, thus reducing the variability induced by distinct exposures. This is an essential issue, as variability may be expected to increase with the increasing complexity of the model, in line with variability in human responses between individuals. Our results, which showed the toxicity of PBS at the ALI in cocultures compared with an incubator control not exposed at the ALI, point to the importance of always including an incubator control in ALI experiments.

The higher complexity of the advanced models makes them more laborious and maybe less applicable for screening purposes. However, this is significantly improved by the efficient experimental design we here present, which also contributes to more robust mechanistical data, as several endpoints are measured from the same insert. One argument that has been used for the application of more complex cell models is that they may increase the sensitivity towards toxic insults. An approach to test the sensitivity towards the induction of SBs is to expose cells with H_2_O_2_ on gels directly before lysis. A striking observation was that the BEAS-2B cells in coculture appeared to be much more sensitive to H_2_O_2_ exposure. Cells in monoculture at ALI showed slightly higher sensitivity to H_2_O_2_ exposure compared to cells in submerged conditions. Also, the endothelial cells showed higher sensitivity to H_2_O_2_ when in coculture with BEAS-2B than as submerged monoculture. The culture media formulation appeared to have no influence on the outcome either in terms of viability or sensitivity towards SBs. The reasons for the higher sensitivity to H_2_O_2_ in coculture are not known and merits further investigations, but it may be related to the above-mentioned interaction between cells in coculture making the DNA more prone to damage.

The higher sensitivity of the BEAS-2B cells in coculture to NM-300K was observed in cell viability by the alamarBlue assay but not on SBs by the comet assay. NM-300K has a high cytotoxic potential and a quite narrow concentration window between non-cytotoxic and cytotoxic effects. During the technical procedure in the sample preparation for comet assay, the cells go through several washing steps, and there are reasons to believe that the more damaged cells are less attached to the insert and can be lost. In the MN assay, the effect of mitomycin-C was most pronounced in the cells from the coculture, which could be an indication that the BEAS-2B cells in coculture may be a more sensitive model for the hazard assessment of genotoxic compounds, and thus a promising advanced model to be adapted for risk assessment based on in vitro data and an IATA approach. The differences in sensitivity between mono- and cocultures, and between the application of lung epithelial A549 cells or bronchial BEAS-2B cell models, for the various endpoints measured emphasizes the importance of carrying out proper characterization of the emerging advanced models, as well as developing robust SOPs. Further, it points to the importance of developing advanced coculture in vitro models, allowing for intercellular signaling, to better mimic tissue organization and enhance the prediction of human hazard.

## 5. Conclusions

An important step in finding the best predictive model for human adverse effects is to increase complexity and thereby obtain more tissue- and organ-like structures, use human cells, characterize the models, and compare responses in different models. This work indicates that the bronchial mono- and coculture models of BEAS-2B and EA.hy926 cells have different sensitivities to NM-300K exposure as measured by cytotoxicity and genotoxicity at DNA and chromosomal levels, and they are different from the alveolar models of A549 and EA.hy926 cells. This is important knowledge to provide more robust reproducible and reliable results, for the further development of advanced 3D respiratory in vitro models relevant to inhalation exposure, and to obtain the most reliable hazard identification and prediction of effects on humans based on non-animal studies. This study provides important knowledge for the further development of advanced 3D respiratory in vitro models for the most reliable hazard identification and prediction of the effects from inhalation on human-based NAMs for NGRA.

## Figures and Tables

**Figure 1 nanomaterials-13-00407-f001:**
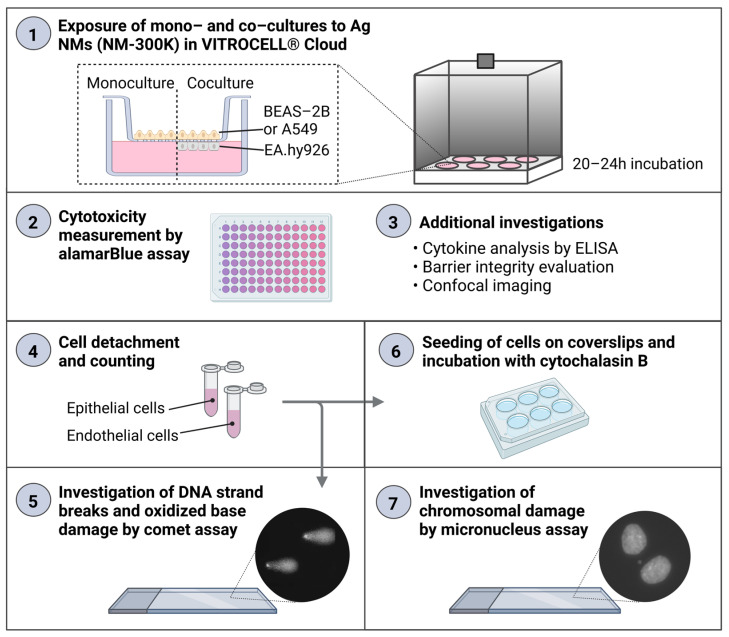
Experimental design for the four different respiratory models exposed at the air–liquid interface (ALI) in the VITROCELL^®^ Cloud system. Created with BioRender.com.

**Figure 2 nanomaterials-13-00407-f002:**
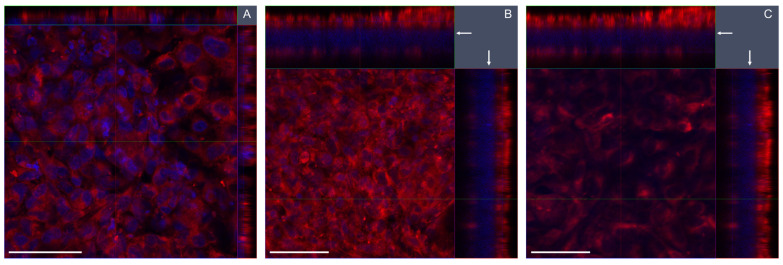
Confocal images of advanced bronchial BEAS-2B models. Z-stack image series (2D x–y view and respective side views) showing the distribution of BEAS-2B and EA.hy926 cells on the opposite sides of a transwell membrane insert (arrow). (**A**) BEAS-2B cells in monocultures (z-stack thickness 12 μm). (**B**) BEAS-2B cells in cocultures (z-stack thickness 52 μm). (**C**) EA.hy 926 cells in cocultures (z-stack thickness 52 μm). Red: cellular membranes stained with Cell Mask red dye; blue: nuclei counterstained with DAPI. Magnification: 40×. Scale bars 50 μm.

**Figure 3 nanomaterials-13-00407-f003:**
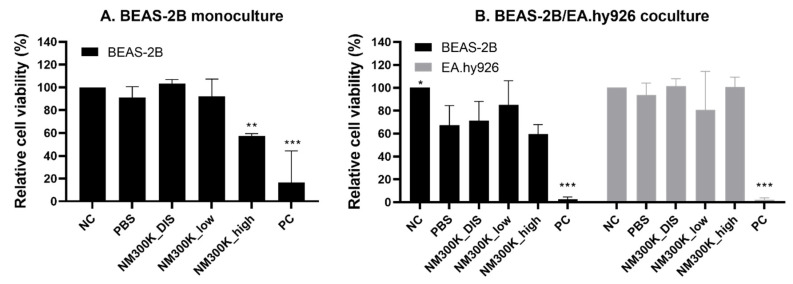
Relative cell viability of BEAS-2B and EA.hy926 cells after exposure to aerosolized NM-300K and control solutions at the air–liquid interface, evaluated by alamarBlue assay. The response of cells in monocultures (**A**) was different compared to cocultures (**B**). Cell viability is presented relative to NC, which is set to 100%. Results are presented as the mean with standard deviation from *n*= 5–7 (**A**) and *n* = 4 (**B**) independent experiments (where the results from each experiment are averaged in the case of two replica inserts). Statistically significant different effects on cell viability compared to control inserts with PBS-exposed cells were analyzed by one-way ANOVA followed by Dunnett´s post-hoc test (* *p* < 0.05, ** *p* < 0.01, *** *p* < 0.001). NC: negative control, PBS: phosphate-buffered saline, NM-300K DIS: dispersant control, NM-300K low: nominal 1 µg/cm^2^, NM-300K high: nominal 10 µg/cm^2^, PC: positive control (chlorpromazine hydrochloride 50–100 µM in basolateral medium for 24 h).

**Figure 4 nanomaterials-13-00407-f004:**
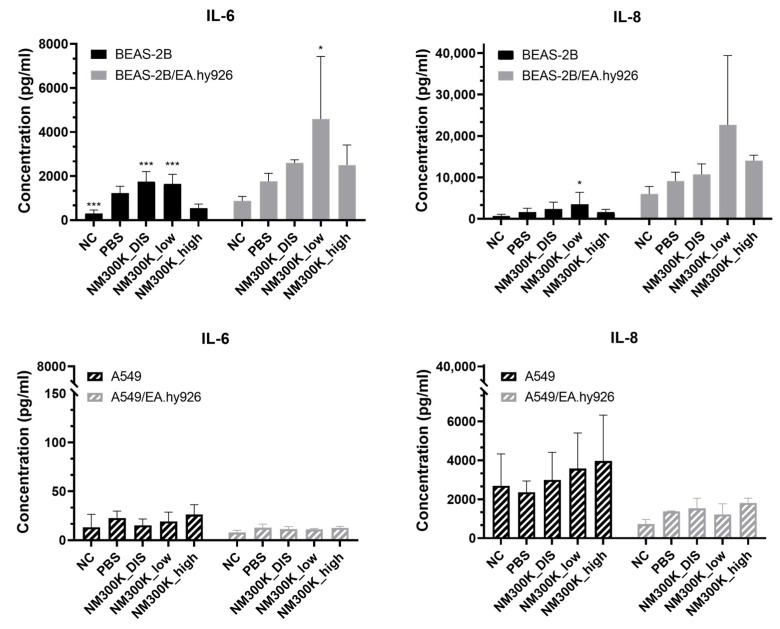
Concentrations of IL-6 and IL-8 in mono- and cocultures of BEAS-2B/EA.hy926 cells (top panel) and A549/EA.hy926 cells (bottom panel) after exposure to aerosolized NM-300K and control solutions at the air–liquid interface, evaluated by ELISA. Results are presented as the mean with standard deviation from single or duplicate inserts from *n* = 2–6 independent experiments (*n* = 6 for BEAS-2B monocultures, *n* = 3 for BEAS-2B/EA.hy926 cocultures and A549 monocultures, and *n* = 2 for A549/EA.hy926 cocultures). Statistically significant different effects on cytokine concentration compared to the negative control inserts with PBS-exposed cells (PBS) were analyzed by one-way ANOVA followed by Dunnett´s post-hoc test (* *p* < 0.05, *** *p* < 0.001). NC: negative control, PBS: phosphate-buffered saline, NM-300K DIS: dispersant control, NM-300K low: nominal 1 µg/cm^2^, NM-300K high: nominal 10 µg/cm^2^.

**Figure 5 nanomaterials-13-00407-f005:**
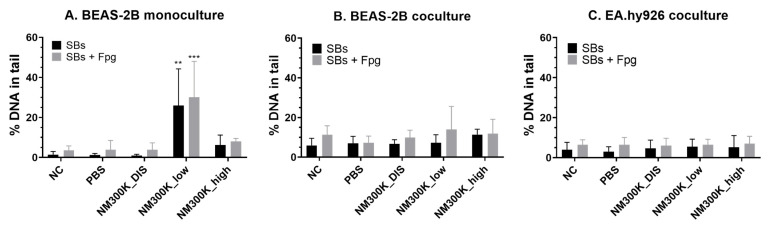
DNA damage by strand breaks (SBs) and oxidized base lesions (Fpg) of BEAS-2B and EA.hy926 cells after exposure to aerosolized NM-300K and control solutions at the air–liquid interface, evaluated by the comet assay. The response of cells in monocultures (**A**) was different compared with cocultures (**B**,**C**). Results are presented as the mean with standard deviation from single or duplicate inserts from *n* = 3–6 (**A**), *n* = 3 (**B**), and *n* = 4 (**C**) independent experiments. Statistically significant different effects on DNA damage compared to control inserts with PBS-exposed cells (PBS) were analyzed by one-way ANOVA followed by Dunnett´s post-hoc test (** *p* < 0.01, *** *p* < 0.001). NC: negative control, PBS: phosphate-buffered saline, NM-300K DIS: dispersant control, NM-300K low: nominal 1 µg/cm^2^, NM-300K high: nominal 10 µg/cm^2^.

**Figure 6 nanomaterials-13-00407-f006:**
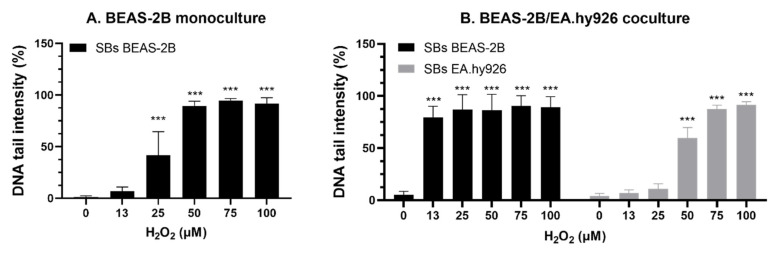
DNA damage by strand breaks (SBs) in BEAS-2B and EA.hy926 cells after exposure to H_2_O_2_ in gels, evaluated by the comet assay. Cells from unexposed inserts of (**A**) BEAS-2B monocultures and (**B**) BEAS-2B/EA.hy926 cocultures were embedded in gels before H_2_O_2_ exposure. Results are presented as the mean with standard deviation from single or duplicate inserts from *n* = 7 (**A**) and *n* = 3 (**B**) independent experiments. Statistically significant different effects of DNA damage compared to negative control cells without H_2_O_2_ exposure (0 µM in PBS) were analyzed by one-way ANOVA followed by Dunnett´s post-hoc test (*** *p* < 0.001). NC: negative control; H_2_O_2_: hydrogen peroxide; PBS: Phosphate-buffered saline.

**Figure 7 nanomaterials-13-00407-f007:**
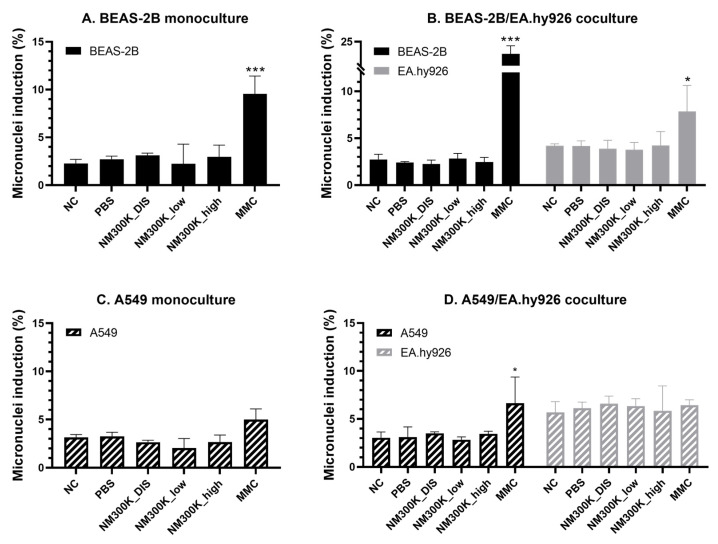
Micronuclei induction in BEAS-2B, EA.hy926 and A549 cells after exposure to aerosolized NM-300K and control solutions at the air–liquid interface, evaluated by the cytokinesis block micronuclei assay. Micronuclei induction was analyzed in cells from BEAS-2B monoculture (**A**), BEAS-2B/EA.hy926 coculture (**B**), A549 monoculture (**C**), and A549/EA.hy926 coculture (**D**). Results are presented as the mean with standard deviation from single or duplicate inserts from *n* = 3 independent experiments. Statistically significant different effects on micronuclei induction compared to control inserts with PBS-exposed cells (PBS) were analyzed by one-way ANOVA followed by Dunnett´s post-hoc test (* *p* < 0.05, *** *p* < 0.001). NC: negative incubator control, PBS: phosphate-buffered saline, NM-300K DIS: dispersant control, NM-300K low: nominal 1 µg/cm^2^, NM-300K high: nominal 10 µg/cm^2^, MMC: mitomycin-C at 0.15 µg/mL in basolateral medium for 24 h.

**Table 1 nanomaterials-13-00407-t001:** Characterization of NM-300K directly after preparation (T0) and after 24 h (T24h). The stock dispersion (10 mg/mL) was used for aerosolized exposure and was diluted in LHC-9 culture medium to make a medium dispersion (141 µg/mL, 100 µg/cm^2^) for submerged exposure. Results are presented as mean ± standard deviation or interval (lower value–higher value) from *n* = 3–15 independent experiments. PDI: polydispersity index, h: hours.

Sample	Time	Z-Ave (nm)	PDI (a.u.)	Main Peak (nm)	*n*
Stock dispersion	T0	149.3 ± 42.5	0.331 ± 0.062	110–265	15
	T24h	129.6 ± 15.4	0.347 ± 0.022	130–147	3
Stock diluted in medium	T0	378.2 ± 109.8	0.393 ± 0.032	148–280	5
	T24h	1838.8 ± 2344.2	0.884 ± 0.739	60–150	5

**Table 2 nanomaterials-13-00407-t002:** Number of live cells, cell density and cell viability from ALI cultures at the end of the cultivation period, evaluated by cell counting with trypan blue staining after detaching the cells from the inserts. Results are presented as mean with standard deviation (SD) from 2–3 independent experiments (*n*) and 2–7 replica culture inserts in each experiment. ALI: air–liquid interface.

ALI Model		Cell Line	Live Cells (×10^6^)	Cell Density (×10^5^/cm^2^)	Viability (%)	*n*
BEAS-2B	Monoculture	BEAS-2B	2.9 ± 0.3	6.8 ± 0.8	95 ± 2	2
	Coculture	BEAS-2B	4.7 ± 0.6	11.1 ± 1.4	97 ± 1	2
		EA.hy926	0.9 ± 0.3	1.6 ± 0.4	83 ± 15	3
A549	Monoculture	A549	2.4 ± 0.7	5.7 ± 1.5	97 ± 2	2
	Coculture	A549	1.4 ± 0.8	3.3 ± 1.9	94 ± 4	2
		EA.hy926	0.8 ± 0.1	1.8 ± 0.1	93 ± 3	2

**Table 3 nanomaterials-13-00407-t003:** Comparison of the barrier integrity of mono- and cocultures of BEAS-2B/EA.hy926 and A549/EA.hy926 cells, by measurement of fluorescein sodium salt and Ag permeation. Results are presented as mean permeation with SD of *n* = 2–3 independent experiments with single or duplicate inserts. Permeation is presented as the basolateral concentration (µM) and as the basolateral concentration relative to the deposited apical concentration (%). Results for A549/EA.hy926 cultures are based on our previous publication [39]. NC: negative control. PBS: phosphate-buffered saline, NM-300K DIS: dispersant control, NM-300K low: nominal 1 µg/cm^2^, NM-300K high: nominal 10 µg/cm^2^.

		Fluorescein or Ag Permeation				
Experiment Type	Treatment	BEAS-2B/EA.hy926 Cultures		A549/EA.hy926 Cultures *		Empty Inserts
		Monoculture	Coculture	Monoculture	Coculture	
Fluorescein (% of deposited apical concentration)	NC	21 ± 2% (*n* = 2) ^a^	69 ± 2% (*n* = 2)	-	-	74 ± 11% (*n* = 2)
	PBS	22 ± 1% (*n* = 2) ^a^	71 ± 7% (*n* = 2)	-	9 ± 5% (*n* = 3) ^a^	
Ag (µM and % of deposited apical concentration) ^b^	NM-300K DIS	0.017 ± 0.003 µM (*n* = 2)	0.026 ± 0.013 µM (*n* = 2)	0.069 ± 0.025 µM (*n* = 2)	0.030 ± 0.017 µM (*n* = 2)	-
	NM-300K low	11.9 ± 1.5 µM 57.1% (*n* = 3) ^c,d^	7.7 ± 1.3 µM 37.1% (*n* = 3) ^e^	1.9 ± 0.6 µM 8.1% (*n* = 2) ^f^	3.3 ± 0.6 µM 14.3% (*n* = 2) ^f^	-
	NM-300K high	11.9 ± 0.4 µM 7.7% (*n* = 3)	8.3 ± 2.2 µM 5.3% (*n* = 3) ^e^	14.5 ± 1.4 µM 8.7% (*n* = 3)	15.4 ± 1.3 µM 9.2% (*n* = 2)	11.40 ± 0.03 µM 7% (*n* = 2)

* Results on A549/EA.hy926 cultures are based on [39]. ^a^ Statistically significant difference from empty inserts, evaluated by one-way ANOVA with post-test Dunnett (*p* < 0.05). ^b^ Ag permeation results were analyzed by one-way ANOVA with post-test Sidak (*p* < 0.05). A total of 16 comparisons were made: between low and high concentration for each model, low and low concentration for BEAS-2B and A549 monocultures, high and high concentration for BEAS-2B and A549 monocultures, low and low concentration for BEAS-2B and A549 cocultures, high and high concentration for BEAS-2B monocultures and A549 cocultures, and high concentration for all models and empty inserts. ^c^ Statistically significant difference from BEAS-2B coculture at the same concentration. ^d^ Statistically significant difference from A549 monoculture at the same concentration. ^e^ Statistically significant difference from A549 coculture at the same concentration. ^f^ Statistically significant difference from the same model at high concentration.

**Table 4 nanomaterials-13-00407-t004:** Comparison of the relative cell viability in bronchial BEAS-2B and alveolar A549 advanced models after aerosol exposure to PBS and NM-300K. Results are presented as the mean relative cell viability (compared to NC, set to 100%) with standard deviation from 4–9 independent experiments (*n*), each with 1–2 replica culture inserts. Statistically significant differences compared to control exposed to PBS were analyzed by one-way ANOVA with Dunnett´s multiple comparisons post-test, and they are indicated by * *p* < 0.05. Data for A549 mono- and cocultures are based on our results from [39]. NC: negative control, PBS: phosphate-buffered saline, NM-300K low: nominal 1 µg/cm^2^, NM-300K high: nominal 10 µg/cm^2^.

	Relative Cell Viability (%)					
	BEAS-2B/EA.hy926 Cultures			A549/EA.hy926 Cultures ^a^		
	Monoculture	Coculture		Monoculture ^a^	Coculture ^a^	
Treatment	BEAS-2B	BEAS-2B	EA.hy926	A549 ^a^	A549 ^a^	EA.hy926 ^a^
**NC**	100 ± 0	100 ± 0 *	100 ± 0	100 ± 0	100 ± 0	100 ± 0
**PBS**	91 ± 9	67 ± 17	94 ± 10	93 ± 23	62 ± 22	69 ± 19
**NM-300K low**	92 ± 15	85 ± 21	81 ± 34	72 ± 13	57 ± 20	67 ± 21
**NM-300K high**	57 ± 2 *	60 ± 8	100 ± 9	59 ± 26	59 ± 29	68 ± 21
** *n* **	5–8	4	4	6–9	4–5	4–5

^a^ Data based on the results from [39].

**Table 5 nanomaterials-13-00407-t005:** DNA damage response after the hydrogen peroxide (H_2_O_2_) exposure of BEAS-2B cells from different culturing conditions. Results are presented as the mean with standard deviation from *n* = 3 independent experiments (except ALI monocultures with *n* = 6), each with 1–2 culture inserts or culture wells per treatment. Medium type is presented as the ratio of medium for BEAS-2B cells (LHC-9) and medium for EA.hy926 cells (DMEM). ALI: air–liquid interface, SBs: strand breaks.

Culture Conditions			H_2_O_2_ Treatment			
			DNA SBs (% DNA in Tail)			EC_50_ (µM)
Culture Type	Cell Line	LHC-9:DMEM	12.5 µM	25 µM	50 µM	
ALI monoculture	BEAS-2B	1:0	7 ± 4	41 ± 23	89 ± 5	28 ± 6
ALI coculture	BEAS-2B	1:1	80 ± 11	87 ± 14	86 ± 15	4 ± 5
	EA.hy926	1:1	7 ± 3	11 ± 5	60 ± 10	45 ± 5
Submerged monoculture	BEAS-2B	1:0	7 ± 3	24 ± 12	84 ± 14	33 ± 6
		1:1	7 ± 3	18 ± 6	77 ± 10	37 ± 6
		0:1	4 ± 3	27 ± 6	85 ± 5	31 ± 1
	EA.hy926	1:0	18 ± 7	20 ± 5	42 ± 17	70 ± 31
		1:1	16 ± 1	19 ± 2	46 ± 19	>100
		0:1	19 ± 3	12 ± 6	34 ± 5	>100

## Data Availability

Data are available from the researchers on request.

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
