# Peer review of "Different Sensitivity of Advanced Bronchial and Alveolar Mono- and Coculture Models for Hazard Assessment of Nanomaterials"

_nanomaterials, 2023, doi:10.3390/nano13030407_

Round 1
Reviewer 1 Report
This paper describes a valuable study evaluating multiple cell lines (monoculture and coculture) for ALI exposure to AgNPs. This is an important topic, but there are unfortunately some limitations that need to be resolved prior to publication.
First, the statistical approach needs to be improved and perhaps a collaboration with a biostatistician is needed. For each comparison, data from multiple experiments are pooled. While this is necessary since only 6 samples can be tested at a time with this Vitrocell system, it does potentially complicate the statistics since variability is included for both replicates within an experiment and also the experiment-to-experiment variability. Therefore, the data for the negative controls for each experiment are linked to test samples for that experiment. Combining all of the data among experiments may expand the variability reducing the potential to identify subtle effects. A more sophisticated statistical model may be useful (e.g., similar to an approach used in a recent paper published in Nanomaterials (doi:10.3390/nano10122369)) on a similar topic. Another overarching statistical question is whether the most important comparison should be to the negative control or to the PBS exposure. Most likely it should be the latter. While this comparison was made according to the statistics section, it was not in the figures. Also, is the data normally distributed? This was not checked.
Second, the scope of the manuscript is too broad and should be sharpened. I strongly encourage the authors to consider deleting sections 3.4 and especially 3.5. Section 3.4 is a somewhat superficial treatment of the topic (2 new experiments performed with only one cell culture system) with limited new experiments conducted and would benefit from a more comprehensive experimental design in a separate paper.
Third, most of the results in this study are negative for the AgNPs, especially if comparisons were made against the solvent (PBS) control. In the limited number of times significant results were observed, the error bars were often huge (greater than 50 % of the mean), lowering confidence in these results. The tone of the manuscript should be adjusted to reflect this.
Lastly, could these results be compared against in vivo data? That would be informative.
Additional comments
Line 285 – Is there a reference to cite here?
Section 2.10 – Why didn’t the authors perform TEER measurements which is the more common approach to test this?
Line 386 – How many aliquots per sample were moved? This should be specified.
Section 2.15 – Was internalization of the NPs into the cells confirmed? This can lead to a false negative result for this assay when testing ENMs.
Line 533 – Was the standard deviation calculated for each experiment or by pooling the results from different experiments?
Table 1 – I think there is a typo here “130-14.” The second number should be larger than the first.
Table 2 and all relevant tables – It would be helpful to provide information about “n” for the total number of replicates too.
Table 3 – I suggest performing statistical comparisons among the conditions
Table 4 – The results from the previous study is unclear. How is one data point for the BEAS-2B from the previous study and only the control data point for the coculture?
Figure 3 - It doesn’t make sense that the NC has no uncertainty. Was there only one NC replicate per run?
Figure 4, bottom – Why is there a cutoff scale bar for the y-axis?
Figure 5, part A – The impact for the NM300K-low only does not make sense since the error bars are huge and there is a lack of a dose-response.
Reviewer 2 Report
Comments
The manuscript by Elje et al, suggest the bronchial mono- and co-culture models of BEAS-2B and EA.hy926 cells have different sensitivity to NM-300K (Nano Ag) exposure measured by cytotoxicity and genotoxicity at DNA and chromosomal levels, and are different from the alveolar models of A549 and EA.hy926 cells. The authors try to develop advanced co-culture in vitro models for better mimic tissue organization and enhanced prediction of human hazards. Although the authors indicated many different outcomes from mono- and co-culture models of cells and attempt to discuss the controversial data in the study, the author's conclusions were not fully supported by their study designs and results. It is weird these results and it is difficult to persuade me to understand the fundamental design of the study.
Major point:
1. Why NM-300K high group did not show any significantly different permeation rate in the monoculture /co-culture mode of barrier integrity experiments (Table 3)? If as the authors mentioned in the discussion section. High concentration of NM-300K may cause agglomeration of the nanoparticles and make them less likely to penetrate the pores of the insert membrane. The authors should bring the evidence to the study.
2. The viability of BEAS 2B cells in cocultures was significantly reduced after aerosol exposure to PBS, with a relative viability of 67 %, compared with the incubator control. This effect of PBS was not seen in the monocultures (Figure 3B). The viability of cells exposed to NM-300K DIS was similar to cells exposed to PBS, in both models. Why the viability of BEAS 2B cells in cocultures was significantly reduced after aerosol exposure to PBS? The authors should explain.
3. In the manuscript, the authors indicated that the experimental design presented in this study, enabling several endpoints to be measured from each insert, allows for higher throughput…However, the number of samples used in the study, only carry out n=2~8. It is not qualifying for high throughput analysis.
Round 2
Reviewer 1 Report
The authors did a sufficient job of responding to the reviewer comments.